# “Guess Who’s Coming to Dinner”: Molecular Tools to Reconstruct *multilocus* Genetic Profiles from Wild Canid Consumption Remains

**DOI:** 10.3390/ani12182428

**Published:** 2022-09-15

**Authors:** Edoardo Velli, Federica Mattucci, Lorenzo Lazzeri, Elena Fabbri, Giada Pacini, Irene Belardi, Nadia Mucci, Romolo Caniglia

**Affiliations:** 1Unit for Conservation Genetics (BIO-CGE), Italian Institute for Environmental Protection and Research (ISPRA), Via Cà Fornacetta 9, 40064 Ozzano dell’Emilia, Italy; 2Research Unit of Behavioural Ecology, Ethology and Wildlife Management, Department of Life Sciences, University of Siena, Via P.A. Mattioli 4, 53100 Siena, Italy

**Keywords:** anthropogenic hybridisation, canid consumption, domestic cat, European wildcat, food habits, *multilocus* genotypes, non-invasive genetic sampling, predation, wolf

## Abstract

**Simple Summary:**

Wolves and European wildcats are two iconic predator species that can live in overlapping ecological contexts and also share their habitats with their domestic free-ranging relatives, increasing the risk of anthropogenic hybridisation and its possible deleterious consequences. By applying a multidisciplinary approach, we morphologically and molecularly analysed the cat remains found in a canid faecal sample collected in a forested area of central Italy. Individual *multilocus* genotypes of both predator and prey were identified turning out to be, respectively, a wolf showing traces of dog ancestry at autosomal microsatellite *loci* and a domestic cat.

**Abstract:**

Non-invasive genetic sampling is a practical tool to monitor pivotal ecological parameters and population dynamic patterns of endangered species. It can be particularly suitable when applied to elusive carnivores such as the Apennine wolf (*Canis lupus italicus*) and the European wildcat (*Felis silvestris silvestris*), which can live in overlapping ecological contexts and sometimes share their habitats with their domestic free-ranging relatives, increasing the risk of anthropogenic hybridisation. In this case study, we exploited all the ecological and genetic information contained in a single biological canid faecal sample, collected in a forested area of central Italy, to detect any sign of trophic interactions between wolves and European wildcats or their domestic counterparts. Firstly, the faecal finding was morphologically examined, showing the presence of felid hair and claw fragment remains. Subsequently, total genomic DNA contained in the hair and claw samples was extracted and genotyped, through a multiple-tube approach, at canid and felid diagnostic panels of microsatellite *loci*. Finally, the obtained individual *multilocus* genotypes were analysed with reference wild and domestic canid and felid populations to assess their correct taxonomic status using Bayesian clustering procedures. Assignment analyses classified the genotype obtained from the endothelial cells present on the hair sample as a wolf with slight signals of dog ancestry, showing a *q*i = 0.954 (C.I. 0.780–1.000) to the wolf cluster, and the genotype obtained from the claw as a domestic cat, showing a *q*i = 0.996 (95% C.I. = 0.982–1.000) to the domestic cat cluster. Our results clearly show how a non-invasive multidisciplinary approach allows the cost-effective identification of both prey and predator genetic profiles and their taxonomic status, contributing to the improvement of our knowledge about feeding habits, predatory dynamics, and anthropogenic hybridisation risk in threatened species.

## 1. Introduction

Trophic relationships within ecosystems are key parameters in wildlife ecological studies [1]. Feeding habits, behaviour and dynamics of animal species, mainly assessed through diet analyses, can provide useful insights into intra- and inter-specific niche specialisation [2]. Moreover, such approaches are particularly suitable for a better comprehension of apex predator ecology, which plays a pivotal role in ecosystem equilibria [3].

During the last decades, diet analyses of vertebrate predators have been extensively conducted through multidisciplinary approaches, mainly based on morphological and molecular identifications of their prey remains contained in non-invasively collected faecal materials [4,5]. In particular, most molecular studies on diet have been focused on a simple taxonomic identification of the consumed species [6,7] using vertebrate broad range markers, such as the cytochrome b and the 16S subunit of the mitochondrial DNA, or applying metabarcoding techniques, to genotype prey DNA contained in predator scats [7,8,9]. However, the recent spread of anthropogenic hybridisation, which originated from the crossing between native and alien species or wild and domestic populations of the same species, requires the application of more powerful molecular tools. The use of panels of highly informative specific markers, such as microsatellite (STR) or single nucleotide polymorphism (SNP) *loci*, permits reconstructing the *multilocus* genetic profiles of both predator and prey when analysing faecal DNA for diet studies. Later on, unknown *multilocus* genotypes can be assigned to well-representative parental populations through statistical procedures to assess species, native populations or signs of admixture. Such data can also provide key information on differences in food dynamics, hunting behaviours and ecological relationships among taxa or between pure and admixed individuals of the same species [10]. This would be particularly useful, especially for two iconic and notoriously elusive Italian carnivores, the Apennine wolf (*Canis lupus italicus*) and the European wildcat (*Felis silvestris silvestris*), which often share their natural habitats and individual territories across the Italian Peninsula, and that can successfully mate with their domestic counterparts, the domestic dog (*C. l. familiaris*) and the domestic cat (*Felis catus*) [11,12].

The wolf and the European wildcat experienced very similar demographic scenarios in Italy, with protracted isolation south of the Alps and recurrent bottlenecks that made them sharply genetically differentiated from any other wolf or wildcat population [13,14]. Nowadays, both species are geographically re-expanding and numerically increasing trough the Peninsula, thanks to legal protection and their ecological plasticity [15,16], but they are still threatened by habitat fragmentation [16,17], accidental or illegal killings [18,19] and by anthropogenic hybridisation [20,21].

Even though distribution ranges and anthropogenic hybridisation, rates in Italy are continuously studied in wolves and European wildcats, especially through non-invasive genetic projects [16,17]. Interspecific relationships between the two species are poorly known, with only a few wolf predations recorded on felids [22,23]. Likewise, differences in behaviour, feeding strategies and diet composition between pure and admixed individuals in both carnivores are still scarcely studied, with only a few available data on a local scale [10].

In this case study, we analysed the DNA contained in the remains of a canid faecal deposition collected in a forested area of central Italy to determine the individual *multilocus* genetic profiles of both the predator and the prey. In particular, we exploited the availability of reliable forensic genetic protocols [24], well-performing panels of canid [25] and felid [26] unlinked autosomal STRs and robust statistical procedures [21] to genotype non-invasive samples, assess their origin and clarify if they had wild, domestic or admixed ancestry.

## 2. Materials and Methods

On 9 February 2020, we collected a canid faecal sample containing the consumption remains of a cat, consisting of hairs and claw fragments, in a wooded area of Tuscany in central Italy. The sample was initially preserved at −20 °C, subsequently stored in an oven at 80 °C for four hours to deactivate possible pathogens, and then frozen again at −20 °C until downstream morphological and molecular analyses.

After separating hairs from claw fragments with water and a 1 mm sieve, a preliminary morphological identification of the hair sample was conducted by observing cuticle and medulla patterns with a 100–400× zoom microscope [27].

Additionally, since stray dogs and cats, as well as wolf-dog and wildcat-domestic cat hybrids, were recorded in the study area, we stored a tuft of hairs (*n* > 10) and the claw fragment, respectively, in a paper envelope to search for possible canid intestinal endothelial cells and in 40 mL of 96% ethanol to be genetically analysed, thus clarifying the taxonomic status of both predator and prey. Total DNA contained in the hair or on its surface (likely containing endothelial intestinal cells of the predator) and in the claw samples was individually extracted using the Blood & Tissue Kit^®^ (Qiagen), following the manufacturer’s instructions.

Each DNA sample was amplified by Polymerase Chain Reaction (PCR) and genotyped through a multiple-tube approach [28] at diagnostic wolf and cat molecular markers. In particular, the diagnostic wolf marker panel included the following: (a) 39 unlinked autosomal microsatellite *loci* (STRs), discriminating among wolves, dogs, and their first three generations of hybrids [12,17]; (b) the *Amelogenin* marker, to molecularly sex the extracted DNA; (c) 4 Y-chromosome STRs (MS34A, MS34B, MSY41A and MS41B Sundqvist et al. 2001) determining the paternal haplotype in male individuals; and (d) a dominant 3-bp deletion at the *β-defensin* CBD103 gene (the *K-locus*) associated with black coat colour. The diagnostic cat marker panel included 29 domestic cat-derived dinucleotide STRs, discriminating among European wildcats, domestic cats, and their first two generations of hybrids [26,29,30].

All PCR reactions were performed in a total volume of 10 µL containing the following: 2 µL of DNA, 5 µL of MasterMix (Qiagen Multiplex Kit), 3 µL of Q-solution (Qiagen Multiplex Kit), 0.15–0.30 µL of primers adjusted to the volume with RNAse-free water. PCR products were analysed in an ABI 3130XL automated sequencer and allele sizes were estimated using Genemapper v.5.0 (Applied Biosystems).

Hair and claw DNA samples were extracted, amplified, and genotyped in three separate rooms dedicated to low-template DNA samples under sterile UV laminar flood hoods. Negative (no biological sample during extraction and no DNA in PCR) and positive (a wolf and a cat DNA sample of good quality and with known genotype) controls were included in each step to check for possible contaminations and correct allelic weights, respectively.

Consensus genotypes, amplification success (AS) and error (allelic dropout, ADO, false alleles, FA) rates were assessed from the four replicates per *locus* performed during the multiple-tube approach, using Gimlet v.1.3.3 [31].

Genotype reliability was calculated by RelioType [32], considering an acceptance value of *R* ≥ 95%.

Reliable genotypes were then assigned to their populations of origin (wolf or dog; European wildcat or domestic cat) through a first explorative Principal Coordinate Analysis (PCoA) using GenAlEx v.6 [33] and Bayesian clustering procedures implemented in Structure v.2.3.4 [34].

Structure was run with three repetitions of 5 × 10^5^ iterations following a burn-in period of 5 × 10^4^ iterations, using the Admixture and Independent Allele Frequencies models [35], and assuming *K* = 2. As reference parental populations, we selected—from the ISPRA *Canis* database—the 39-STR genotypes of 190 unrelated wild individuals belonging to the Italian wolf population and 89 wolf-sized dogs living in rural areas [21]. Based on previous analyses performed on simulated wild, domestic, and hybrid genotypes, we assigned the unknown canid genotype to the Italian wolf population if its wolf membership proportion was *q*i ≥ 0.955, to the dog population if *q*i < 0.2, otherwise it was considered as admixed for intermediate (0.2 < *q*i < 0.954) values (see Caniglia et al. [21] for further information about reference population choice and threshold selection).

A similar approach was used for the felid genotype, using as reference parental populations the 29-STR genotypes of 48 unrelated European wildcats, representative of the species distribution range in the central Apennines, and 65 free-ranging domestic cats, selected from the ISPRA *Felis* database [26]. However, when using another specific molecular marker set, such as the mentioned 29 domestic cat-derived STR panel [26,29,30], simulated genotype analyses suggested the application of different dedicated detection thresholds, assigning the unknown felid genotype to the European wildcat population if its wildcat membership proportion was *q*i ≥ 0.8, to the domestic cat if *q*i < 0.2, whereas it was considered as admixed for intermediate (0.2 < *q*i < 0.799) values [11].

## 3. Results

The analysis of cuticle and medulla patterns morphologically classified the hair sample as belonging to *Felis* sp., suggesting it was a domestic cat, though genetic analyses on the same specimen returned a reliable *multilocus* genotype (*R* > 95%) only at the wolf microsatellite panel with an average rate of missing data of 0.154. Conversely, the claw DNA sample yielded a reliable genotype (*R* > 95%) only at the felid microsatellite panel with an average rate of missing data of 0.315. Mean rates of AS, ADO, and FA across loci were 0.846, 0, and 0 (standard errors: 0.056, 0, 0) for the hair sample and 0.638, 0.049 and 0.009 (standard errors: 0.083, 0.03, 0.009) for the claw sample, respectively. In the PCoA including canid reference populations, the hair genotype plotted marginally to reference wolves (Figure 1a), whereas in the PCoA including felid reference populations the claw genotype completely overlapped with reference domestic cats (Figure 1b). Assignment procedures clearly supported the outcomes from the PCoA, classifying the canid genotype obtained from the hair sample as belonging to an individual with slight genetic signs of dog ancestry, showing a proportion of posterior probability to belong to the wolf cluster *q*i = 0.954 (95% confidential interval C.I. = 0.78–1, Figure 1c). The *Amelogenin* analysis suggested it was a male (heterozygous genotype) individual with a Y-chromosome haplotype typical of the Italian wolf population [17], whereas the absence of the K-*locus* deletion theoretically indicated an animal with a wild-type coat colour. The claw genotype was unambiguously assigned to the domestic cat group with a proportion of posterior probability *q*i = 0.996 (95% C.I. = 0.982–1, Figure 1d).

## 4. Discussion

Non-invasive genetic sampling can provide useful insights into trophic relationships within ecosystems, clarifying fundamental ecological aspects such as feeding habits, predation behaviour, and dynamics of animal species, thus contributing to designing sound and long-term conservation strategies [36,37,38]. This approach is particularly when applied to elusive and threatened large or meso-carnivores such as the wolf and the European wildcat [23,39]. These two iconic species can somewhat share their ecological niches and sometimes their territories could overlap those of their domestic free-ranging relatives, increasing the risk of anthropogenic hybridisation and its possible deleterious consequences [40]. Such phenomenon, if widely spread, could undermine the gene pool integrity of wild ancestors through the introgression of domestic artificially selected genetic variants, which might potentially affect morphological, physiological, and behavioural traits of natural populations [13,41].

In this case study, we used the ecological, morphological, and genetic information contained in a biological canid faecal sample to detect any sign of trophic interactions between wolf and European wildcat individuals or their domestic counterparts. Though traditional molecular diet analyses are not able to identify with certainty the taxonomic status of the involved *taxa* below the specific level [7], we overcame this limitation by applying a multidisciplinary detection approach. 

Firstly, we conducted traditional morphological analyses, which revealed the presence of felid remains in the collected canid faecal sample. Then, we used reliable molecular tools and highly discriminant canid and felid STR panels [12,26], which allowed us to reconstruct both the predator and prey genetic profiles. Finally, we exploited the availability of well-represented wild and domestic canid and felid reference populations to identify their correct taxonomic status at the subspecies level through Bayesian clustering analyses. 

Interestingly, despite the initial water treatment of the faecal sample to separate its content, the few remaining canid intestinal cells on the hair sample allowed us to fully reconstruct the predator genetic profile with very high amplification success rates and no sign of ADO or FA across *loci*. Such a profile resulted in a wolf with traces of dog ancestry at autosomal microsatellite *loci*, consistently with the presence of wolf-dog admixture cases, repeatedly documented in this region using both video-trapping and genetic tools [13,42]. We were also able to reconstruct the genetic profile of the individual to which the claw belonged but with lower amplification success and higher error rates, which were, however, consistent with those reported in other non-invasive felid studies [16,20,43], probably due to the partial intestinal digestion the claw was exposed to. Such a felid profile resulted in a domestic cat and assessment of the problematic presence of free-ranging cat individuals in rural areas. This might increase the risk of anthropogenic admixture with wild animals as well as of predation on small vertebrates, thus bearing negative consequences for the conservation of biodiversity.

Sporadic cat consumption by wolves has been reported [44] and our findings might suggest a trophic interaction between admixed and domestic individuals into the wild. However, the analysis of food remains from scats cannot rule out the *post-mortem* consumption of a cat carcass. Although our results were obtained from a single case study, future genetic identifications from faecal samples combined with accurate detections of the consumed prey, if extensively applied, could provide additional useful data about possible shifts in food habits and habitat use between wild predators and their domestic counterparts. This is particularly useful in areas with a documented presence of wild and domestic overlapping [10,45], though to date significant different foraging strategies between wolf-dog admixed individuals and wolves have not been documented in the few available studies [10], and no similar data are currently reported for European wildcats and domestic cats.

Our multidisciplinary approach could also be applied to support future studies on possible behavioural alterations between wolves and wolf-dog hybrids in predatory strategies, contributing to detect the presence of admixture on one of their main preys, the wild boar, which is also facing anthropogenic hybridisation with the domestic pig [46,47]. However, results from this case study, although promising, are preliminary and should be confirmed by further studies based on larger sample sizes to statistically support the reliability of the applied method.

Our methodological procedures could be further improved by using highly performing markers to genotype non-invasively collected samples such as metabarcoding or highly informative and easily inter-lab comparable SNP panels, in cases of canid or felid predation on wildlife or other domestic pets to establish the real taxonomic status of the predators through a finer scale admixture identification and to provide a more reliable assessment of their potential impact on other threatened species [20,47,48,49].

## 5. Conclusions

Our results, although limited to the analysis of a single biological sample, clearly show how a multidisciplinary approach allows the cost-effective identification of both predator and prey *multilocus* genetic profiles and the accurate detection of their taxonomic status from non-invasive DNA. This will contribute to improve our knowledge about feeding habits, predatory dynamics, and anthropogenic hybridisation risk in threatened species, thus collecting information which is key to plan adequate conservation management actions.

## Figures and Tables

**Figure 1 animals-12-02428-f001:**
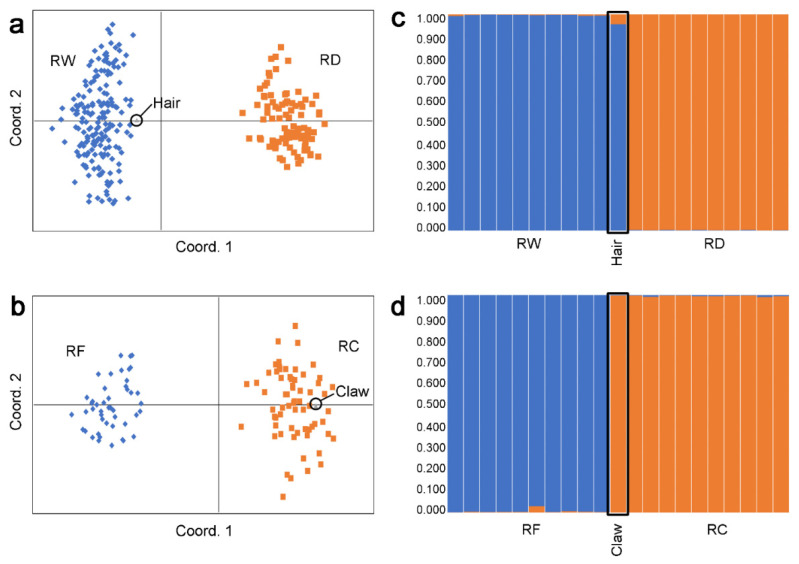
Multivariate and Bayesian assignment of *multilocus* genotypes obtained from the hair and claw samples to their respective belonging populations. Left side: graphical plotting of the Principal Coordinate Analysis performed in GenAlEx assigning (**a**) the hair sample genotype (grey triangle) to the cluster of the reference wolf (RW, blue) or dog (RD, orange) populations; (**b**) the claw sample genotype (grey triangle) to the cluster of the reference wildcat (RF, blue) or domestic cat (RC, orange) populations. Right side: Bayesian clustering histograms produced by Structure assuming *K* = 2 clusters and with the “admixture” and “I” models of the (**c**) hair genotype determined at 39 microsatellite *loci* and assigned to the reference wolf (RW, blue) or dog (RD, orange) populations and (**d**) claw genotype determined at 29 microsatellite *loci* and assigned to the reference wildcat (RF, blue) or domestic cat (RC, orange) populations. Each individual is represented by a vertical bar fragmented into the two-coloured sections, according to their proportion of membership to the reference population clusters. For a better readability, only ten genotypes for each reference population were shown as bar plots obtained from Bayesian assignment procedures.

## Data Availability

Data presented in this study are available on request.

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
