# Peer review of "“Guess Who’s Coming to Dinner”: Molecular Tools to Reconstruct multilocus Genetic Profiles from Wild Canid Consumption Remains"

_animals, 2022, doi:10.3390/ani12182428_

Round 1

Reviewer 1 Report

In the ms “ Guess Who’s Coming to Dinner”: molecular tools to reconstruct multilocus genetic profiles from wild canid consumption remains” authors analyse one fecal sample with molecular methods in order to obtain information about the genetic profiles of the predator and the prey. The objective alleged by authors is obtain information about interactions between wolves (Canis lupus) and wildcat (Felis silvestris silvestris). However, one sample is not enough to address this. In fact, authors only have the confirmation of a predation event of a wolf over a domestic cat. Although the methodology is well developed regarding molecular procedures, the ecological interactions remains unsolved. In consequence, the conclusions are not supported by results. We also could interpret this work as a methodological one, but in my opinion this is not a new method, as it is extensively used, including studies with a wider number of samples.

I recommend authors to re-think the work, maybe by enlarge the number of samples. Allowing a later ecological analysis. Other option is to develop a review with wolf and felid predator-prey interactions, and its possible consequences in ecosystems.

Specific comentaries:

Lines 249-255: This is extensively used nowadays in order to confirm the identity of the predator, like per example in the case of wolves in spain, where sheperds claim economic aids to governement when their herd suffer a confirmed wolf attack.

Author Response

Response to Reviewer 1 Comments

Dear Reviewer,

Thank you very much for evaluating our manuscript titled “Guess Who’s Coming to Dinner: molecular tools to reconstruct multilocus genetic profiles from wild canid consumption remains”. Please find enclosed our answers to your comments and considerations.

Question (Q1) - In the ms “Guess Who’s Coming to Dinner”: molecular tools to reconstruct multilocus genetic profiles from wild canid consumption remains” authors analyse one faecal sample with molecular methods in order to obtain information about the genetic profiles of the predator and the prey. The objective alleged by authors is obtain information about interactions between wolves (Canis lupus) and wildcat (Felis silvestris silvestris). However, one sample is not enough to address this. In fact, authors only have the confirmation of a predation event of a wolf over a domestic cat. Although the methodology is well developed regarding molecular procedures, the ecological interactions remain unsolved. In consequence, the conclusions are not supported by results. We also could interpret this work as a methodological one, but in my opinion, this is not a new method, as it is extensively used, including studies with a wider number of samples.

I recommend authors to re-think the work, maybe by enlarge the number of samples. Allowing a later ecological analysis. Other option is to develop a review with wolf and felid predator-prey interactions, and its possible consequences in ecosystems.

Reply (R1) - Thank you very much for your comments and considerations on our manuscript. We are conscious that results obtained from one sample do not have any statistical power to draw any conclusion on ecological relationships among taxa. However, the primary aim of our study was not to fill the gap of information about interactions between wolves and wildcats in Italy, but to “determine the individual multilocus genetic profiles of both the predator and the prey” from the collected sample, “assess their origin and clarify if they had wild, domestic or admixed ancestry”. We additionally agree with Reviewer #1 that our approach is not a new method since we used classic microsatellite panels for multilocus genotype reconstructions from non-invasively collected samples. However, the novelty of our study consists in applying marker panels commonly used in wildlife non-invasive genetic studies and hybrid detection to obtain a high level of identification of both prey and predator and of discrimination between closed related taxa, like wild and domestic relatives, from a single faecal sample, which is not usually achieved in molecular diet studies. Thus, our goal was to exploit a single datum to propose a valid approach that, if applied to broader non-invasive genetic monitoring campaigns, could help in collecting data to clarify some aspects of trophic interaction among taxa. In this perspective, we think that developing a review on wolf and felid predator-prey interactions, although it would be an extremely interesting topic, would overlook the meaning of our limited but clear findings on which we focused in this case study. We modified the manuscript trying to better explicit our intents and aims and adding cautions on the limited sample size, highlighting how this study could represent just a pilot example of such approach and would need a broader application to confirm its reliability and provide ecologically meaningful conclusions.

Specific commentaries:

Q2 - Lines 249-255: This is extensively used nowadays in order to confirm the identity of the predator, like per example in the case of wolves in Spain, where shepherds claim economic aids to government when their herd suffer a confirmed wolf attack.

R2 - Thank you. We agree that predator identification from depredated livestock carcasses is a well-established procedure. We realized that our sentence could be actually unclear. We better specified that a fine-scale identification of the prey (i.e., a domestic or a wild animal) from a predator scat could help in detecting hidden events of possible pet depredation and simultaneously clarify the taxonomic status of the predator. We modified the sentence accordingly.

Reviewer 2 Report

The ms "Guess Who’s Coming to Dinner”: molecular tools to reconstruct multilocus genetic profiles from wild canid consumption remains" describes a case study where non-invasive genetic techniques and a multidisciplinary approach is applied to analyze a canid faecal sample collected in central Italy. Although it presents only one case study performed on a single sample, the work highlights the potential of multilocus genotyping to identify prey and predator genetic profiles.

Some comments and suggestions are listed below:

1. Summary: the second sentence might be split into two shorter and clearer sentences

"Applying a multidisciplinary approach, we morphologically and molecularly analysed the cat remains found in a canid faecal sample collected in a forested area of central Italy. Individual multilocus genotypes of both predator and prey were identified and resulted to be, respectively, a wolf showing traces of dog ancestry at autosomal microsatellite loci, and a domestic cat."

2.  "European wildcat" might be cut from Keywords

3. Introduction

Line 48: Correct to "... for a better comprehension of apex predators ecology, which plays a pivotal role in ecosystem equilibria."

Line 50: "During the last decades, diet analyses of vertebrate predators have been..."

Line 80: "Despite distribution ranges and anthropogenic hybridization rates in Italy are continuously studied in wolves and European wildcats,..."

Line 83: "Differences in behaviour, feeding strategies and diet composition between pure and admixed individuals in both carnivores are still poorly studied, with only a few data available on a local scale."

4. M&M

Line 107: Cut "Therefore" and start sentence directly: "Total DNA from the hair or on its surface..."

Line 151: Explain and justify briefly, although Reference is present, why different thresholds (qi) were chosen for assigning canid and felid genotypes (for instance why qi ≥ 0.800 was chosen for wildcat/domestic cat membership proportion).

5. Results

Line 169: Correct clew to claw

Figure 1: Graphical plotting and bar plots might be bigger in the page.

6. Discussion

Try not to repeat the exact same sentences between Abstract, Introduction and Discussion.

Line 199: Start sentence directly: "These two iconic species can live..."

Line 221: "fully reconstruct the predator genetic profile with very high amplification success..."

Line 225: "We were also able to reconstruct the genetic profile..."

Line 228: "the claw was exposed to."

Line 229: "Such a felid profile resulted in a domestic cat, addressing the problematic presence of free-ranging cat individuals in rural areas..."

Line 236-243: The sentence is too long, split into two.

Line 245: .."possible behavioural alterations among wolves and wolf-dog hybrids.."

Line 246: .."contributing to detect the presence of admixture.."

Line 251-252: "using highly performing markers to genotype non-invasively collected samples"

Cut "that could be analysed, for instance, in Fluidigm platforms," to shorten the sentence. 

Author Response

Response to Reviewer 2 Comments

Dear Reviewer,

Thank you very much for evaluating our manuscript titled “Guess Who’s Coming to Dinner: molecular tools to reconstruct multilocus genetic profiles from wild canid consumption remains”. Please find enclosed our answers to your comments and considerations.

Question (Q0) - The ms "Guess Who’s Coming to Dinner”: molecular tools to reconstruct multilocus genetic profiles from wild canid consumption remains" describes a case study where non-invasive genetic techniques and a multidisciplinary approach is applied to analyse a canid faecal sample collected in central Italy. Although it presents only one case study performed on a single sample, the work highlights the potential of multilocus genotyping to identify prey and predator genetic profiles.

Reply (R0) - Thank you very much for your detailed review. We followed almost all the suggestions you gave and answered your comments. We think that your considerations significantly improved the quality of the manuscript.

Some comments and suggestions are listed below:

Q1 - Summary: the second sentence might be split into two shorter and clearer sentences.

"Applying a multidisciplinary approach, we morphologically and molecularly analysed the cat remains found in a canid faecal sample collected in a forested area of central Italy. Individual multilocus genotypes of both predator and prey were identified and resulted to be, respectively, a wolf showing traces of dog ancestry at autosomal microsatellite loci, and a domestic cat."

R1 - Thank you. We modified it as suggested.

Q2 - "European wildcat" might be cut from Keywords

R2 - Thank you, we understand your point. Nonetheless, we think that this case study could be useful also to researchers working on European wildcats since this approach could detect traces of hybridization between wild and domestic relatives (that represents a primary threat also for wildcat populations) and could be applied in overlapping ecological contexts.

Introduction

Q3 - Line 48: Correct to "... for a better comprehension of apex predators ecology, which plays a pivotal role in ecosystem equilibria."

R3 - Thank you. We modified it as suggested.

Q4 - Line 50: "During the last decades, diet analyses of vertebrate predators have been..."

R4 - Thank you. We modified it as suggested.

Q5 - Line 80: "Despite distribution ranges and anthropogenic hybridization rates in Italy are continuously studied in wolves and European wildcats,..."

R5 - Thank you. We modified it as suggested.

Q6 - Line 83: "Differences in behaviour, feeding strategies and diet composition between pure and admixed individuals in both carnivores are still poorly studied, with only a few data available on a local scale."

R6 - Thank you. We modified it as suggested.

  1. M&M

Q7 - Line 107: Cut "Therefore" and start sentence directly: "Total DNA from the hair or on its surface..."

R7 - Thank you. We modified it as suggested.

Q8 - Line 151: Explain and justify briefly, although Reference is present, why different thresholds (qi) were chosen for assigning canid and felid genotypes (for instance why qi ≥ 0.800 was chosen for wildcat/domestic cat membership proportion).

R8 - Thank you. We adjusted this section clarifying that dedicated analyses on simulated wild, domestic and hybrid individuals were performed on both taxa. Moreover, we justified the different threshold values chosen by dealing with different species and different marker sets.

  1. Results

Q9 - Line 169: Correct clew to claw

R9 - Thank you. We modified it as suggested.

Q10 - Figure 1: Graphical plotting and bar plots might be bigger in the page.

R10 - We followed the instruction for authors in preparing the draft, using dedicated templates. We think that the figure dimensions will be adjusted by the journal graphic office after acceptance.

  1. Discussion

Q11 - Try not to repeat the exact same sentences between Abstract, Introduction and Discussion.

R11 - Thank you. We modified the sentences as suggested.

Q12 - Line 199: Start sentence directly: "These two iconic species can live..."

R12 - Thank you. We modified the paragraph in other sections however we think that in this case the adverb “Indeed” could reinforce the sentence.

Q13 - Line 221: "fully reconstruct the predator genetic profile with very high amplification success..."

R13 - Thank you. We modified it as suggested.

Q14 - Line 225: "We were also able to reconstruct the genetic profile..."

R14 - Thank you. We modified it as suggested.

Q15 - Line 228: "the claw was exposed to."

R15 - Thank you. We modified it as suggested.

Q16 - Line 229: "Such a felid profile resulted in a domestic cat, addressing the problematic presence of free-ranging cat individuals in rural areas..."

R16 - Thank you. We modified it as suggested.

Q17 - Line 236-243: The sentence is too long, split into two.

R17 - Thank you. We split the sentence as suggested.

Q18 - Line 245: .."possible behavioural alterations among wolves and wolf-dog hybrids.."

R18 - Thank you. “Wolf and wolf-dog hybrids” are two distinct categories, therefore we think it could be appropriate to use the preposition “between” instead of “among”.

Q19 - Line 246: .."contributing to detect the presence of admixture.."

R19 - Thank you. We modified it as suggested.

Q20 - Line 251-252: "using highly performing markers to genotype non-invasively collected samples".

Cut "that could be analysed, for instance, in Fluidigm platforms," to shorten the sentence.

R20 - Thank you. We modified it as suggested.

Round 2

Reviewer 1 Report

The authors have made an effort to better contextualize the manuscript, indicating the real contribution of it. Regardless of the interest or relevance, for me this manuscript is now suitable for publication